# Measurement of charges and chemical bonding in a cryo-EM structure

Saori Maki-Yonekura [1,5], Keisuke Kawakami [1,5], Kiyofumi Takaba [1], Tasuku Hamaguchi[1,4] & Koji Yonekura [1,2,3 ✉]

Hydrogen bonding, bond polarity, and charges in protein molecules play critical roles in the stabilization of protein structures, as well as affecting their functions such as enzymatic catalysis, electron transfer, and ligand binding. These effects can potentially be measured in Coulomb potentials using cryogenic electron microscopy (cryo-EM). We here present charges and bond properties of hydrogen in a sub-1.2 Å resolution structure of a protein complex, apoferritin, by single-particle cryo-EM. A weighted difference map reveals positive densities for most hydrogen atoms in the core region of the complex, while negative densities around acidic amino-acid side chains are likely related to negative charges. The former positive densities identify the amino- and oxo-termini of asparagine and glutamine side chains. The latter observations were verified by spatial-resolution selection and a dose-dependent frame series. The average position of the hydrogen densities depends on the parent bonded-atom type, and this is validated by the estimated level of the standard uncertainties in the bond lengths.

[1] Biostructural Mechanism Laboratory, RIKEN SPring-8 Center, 1-1-1 Kouto, Sayo, Hyogo 679-5148, Japan. [2] Institute of Multidisciplinary Research for Advanced Materials, Tohoku University, 2-1-1 Katahira, Aoba-ku, Sendai 980-8577, Japan. [3] Advanced Electron Microscope Development Unit, RIKEN-JEOL Collaboration Center, RIKEN Baton Zone Program, 1-1-1 Kouto, Sayo, Hyogo 679-5148, Japan. [4] Present address: Institute of Multidisciplinary Research for Advanced Materials, Tohoku University, 2-1-1 Katahira, Aoba-ku, Sendai 980-8577, Japan. [5] These authors contributed equally: Saori Maki-Yonekura, Keisuke Kawakami. ✉email: yone@spring8.or.jp

t is now recognized that the excellent coherence of an electron beam is a significant factor in improving spatial resolution with single-particle cryogenic electron microscopy (cryo-EM)[1–4]. The cold field emission (CFE) gun and monochromator produce a high temporal-coherence beam, which retains better signals beyond 2 Å (Figs. 2 and 3 of Hamaguchi et al.[1]; Supplementary Fig. 1). Indeed, imaging with the CFE beam or through the monochromator yielded single-particle reconstructions of the protein complex apoferritin, which is often used as the benchmark for single-particle analysis (SPA), at 1.25 Å and 1.22 Å resolutions[2,3].

Thereby, SPA reached a so-called atomic resolution, at which individual atoms appear as spheres[2,3], and difference maps even show signals of hydrogen atoms[3,5]. Hydrogen atoms and their bonding play important roles in the stabilization of protein structures and in functions such as enzymatic catalysis, electron transfer, and ligand binding[6–8], but identification of hydrogen atoms is difficult due to their low visibility by X-rays[5,8] and electrons[9–11]. Thus, this is a big step with single-particle cryo-EM.

Further challenges would include what and how chemical information beyond atomic coordinates could be extracted from single-particle cryo-EM data. X-rays are scattered by the electrons around atoms, while electrons by Coulomb potential are in their path. Reflecting this, electron scattering factors vary considerably between neutral and charged atoms, particularly at low resolution[12–14]. This feature has been exploited to measure charge distributions in covalent crystals of inorganic materials from convergent-beam electron diffraction (CBED) patterns[15,16] and to investigate charged states at functional sites in thin crystals of proteins[12] and small organic compounds[11] by rotation 3D electron diffraction with a parallel beam (3D ED). However, different from those techniques, single-particle cryo-EM by bright-field electron imaging is affected by many factors. These include phase errors caused by intensity variations and beam tilts from the axial coma-free axis, modification and decay of amplitudes by the contrast transfer function (CTF), and errors in Euler angular assignment for 3D reconstruction. Progresses in hardware, theories, and software have reduced these difficulties, and we believe that single-particle cryo-EM now could have the potential to measure more detailed properties, including information related to charges. This will expand application targets for the investigation of experimental Coulomb potentials. Of course, charges are critically important for the function of macromolecules. But, studies of charge distributions inside proteins have been done mainly by ultra-high resolution X-ray crystallography[17–19] that are limited to samples with superb crystallinity. The application of 3D ED still has unsolved tasks, such as attributions of partial charges and treatment of charged scattering factors[11–14].

Here, we report a sub-1.2 Å resolution structure of apoferritin reconstructed from images collected with the CFE beam. The visibility of hydrogen atoms in the cryo-EM map is improved, which now allows identification between the amino- and oxo-termini of asparagine and glutamine side chains. We then carried out extended analyses to measure more detailed chemical properties inside the protein complex.

## Results and discussion
**Structure analysis.** More than 2,000,000 particles from ~7900 image stacks (Supplementary Table 1; Supplementary Fig. 2) gave a 1.21 Å resolution map, and the final resolution reached 1.19 Å (Supplementary Table 1; Supplementary Fig. 3) by rescaling to finer sampling with the super-resolution pixels of the K3 detector. This dataset is named A (Supplementary Figs. 2 and 3). Beam tilts during data collection were small in this dataset, and the data quality of the data is further discussed in Supplementary Discussion. The

estimated Rosenthal–Henderson B-factor (Supplementary Fig. 3)[20] is comparable to the previous reconstructions[2,3]. We used simple holey carbon film on copper grids for sample support, but the use of gold grids may further improve the resolution[21].

**Structure features.** The cryo-EM map resolves separate densities for individual atoms, particularly in the core region of the protein complex (Fig. 1 and Supplementary Fig. 4), while side chains at the surface of the apoferritin spherical shell seem rather flexible with some residues showing multi-conformations (Supplementary Fig. 5). These features have been observed in previous structure analyses[2,3]. The high quality of the cryo-EM map allowed us to build an atomic model unambiguously, and the model metrics are good (Supplementary Table 1).

We then calculated a weighted $F_o$–$F_c$ difference map between the experimental data and the model omitting hydrogen atoms by Servalcat[5]. The weighted difference map reveals many densities corresponding to hydrogen atoms. The number of hydrogen atoms associated with the protein model is 905, which corresponds to ~70% of total possible hydrogen atoms, after picking the atoms at a density level of ≥2σ and selecting manually based on their riding positions (see "Methods"). In the protein core, most of the hydrogen atoms are identified for both the main chain and side chains. Representative structures are shown in Fig. 1b, c, and Supplementary Fig. 4a, e. Structures around a tyrosine (Tyr 168) and phenylalanine (Phe 51) clearly exhibit all the hydrogen atoms. The hydrogen density in the hydroxyl group of Tyr 168 is bent towards an oxygen atom in a neighboring residue for hydrogen bonding (Fig. 1b). Some water molecules also show resolved hydrogen densities (Fig. 1b; Supplementary Fig. 4a), and we identified six water molecules with two separate hydrogen densities, which contribute to the formation of a water cluster through hydrogen bonding (Supplementary Fig. 4a). One of them, shown in Supplementary Fig. 4b, fits well to the configuration of a theoretical water model. A water molecule (marked with '*') at the lower left of W5 shows no clear hydrogen densities, probably because the water resides in multi-dispositions at this site due to fewer unique partners for hydrogen bonding.

Typical maps are presented for several amino acid types in Fig. 1c. Reflecting the pH 7.5 of the sample solution, acidic amino acid residues appear to be de-protonated. The amino- and oxo-termini in asparagine and glutamine can be clearly identified by the presence or absence of hydrogen densities (Fig. 1c).

**Attribution of negative densities.** The carboxyl-termini of the side chains in the aspartate and glutamate are well resolved (Fig. 1c), although the side chains of aspartate and glutamate are known to be particularly susceptible to radiation and sometimes disappear by exposure to a high X-ray dose[22]. In the protein structures obtained by 3D ED, we observed decreased densities for negatively charged atoms[12] since electron scattering factors of anions become negative or nearly zero at lower spatial resolutions[12–14]. A cut of lower-resolution data recovered some densities for the side chains of aspartate residues[12]. In single-particle analysis, amplitude information is less accurate due to the effects of CTF, and the standard post-process procedure sharpens the map. This amplification of high-resolution parts and/or amplitude modification appear to maintain the densities of acidic amino acid side chains in the experimental maps (Fig. 1c). Notable damage, such as decarboxylation[22] is not observed in Glu17 and Asp 17. However, in the weighted difference map, there is only a small negative density at level ≤−4σ near the terminal oxygen atom of Gln 23 and no around Asn 25 but larger and more dispersed negative densities at the same level around

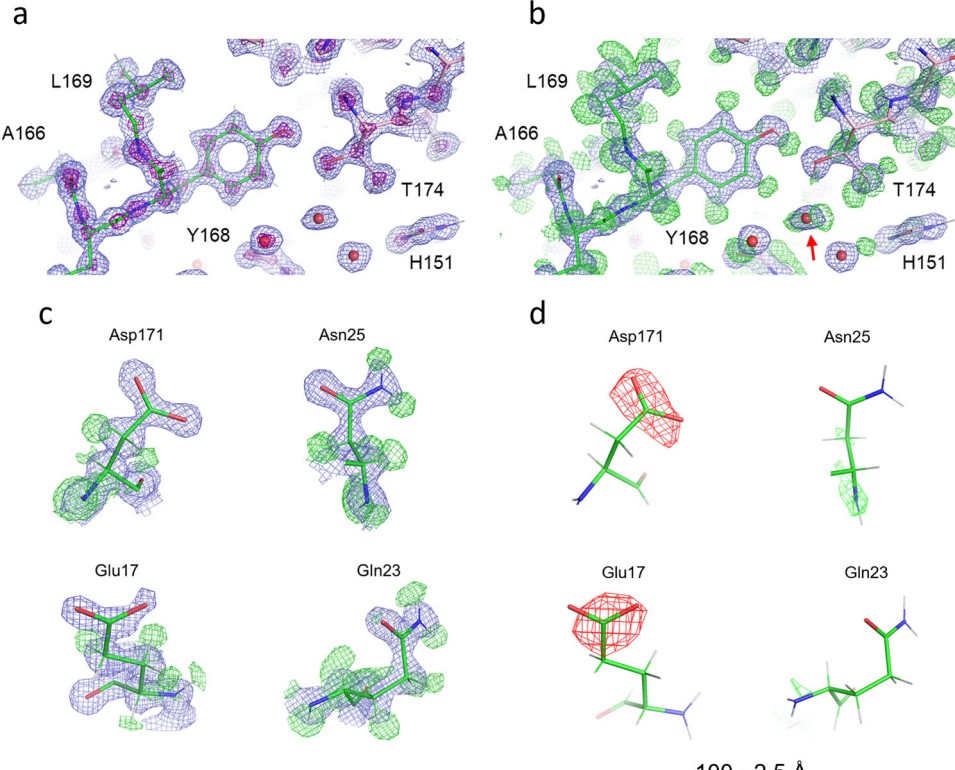

100 - 2.5 Å

**Fig. 1 Structure details of apoferritin. a, b** View around a tyrosine (Tyr 168) overlaid with the atomic model refined in this study. Experimental densities are shown in blue (**a**, **b**) and purple (**a**), and positive densities in a weighted difference map between the experimental data and the hydrogen-omitting model in green (**b**). An arrow in **b** indicates a water molecule (numbered 320 in the PDB) with two hydrogen densities in green. **c** Zoom-up views of single amino-acid residues, aspartate (Asp 171), asparagine (Asn 25), glutamate (Glu 17), and glutamine (Gln 23) with experimental densities in blue and positive densities in the weighted difference map in green. **d** The same as in **c**, but with positive and negative densities in green and red, respectively, in the weighted difference map calculated by limiting the data to 2.5 Å resolution. Blue nets in **a–c** and purple nets in **a** are displayed at density levels of 2σ and 7σ, respectively. Green nets in b and c show differences at 3σ, and green and red nets in d are 4 and −4σ, respectively.

the side chains of Asp 171 and Glu 17 (Fig. 2a). Here, we selected these residues with estimated temperature factors (B-factors) of terminal oxygen atoms <25 Å$^2$ except for Glu 17 and used model structure factors consisting of all neutral atoms for calculating the weighted difference map. Removal of the data at resolutions higher than 2.5 Å reveals prominent negative densities only on the acidic side chains (Figs. 1d and 2b). Such strong negative densities gradually decrease as lower-resolution data are omitted (Fig. 2c). These observations are consistent with the fact that electron scattering factors vary considerably between neutral and charged atoms at low resolution but match at resolutions better than ~2.5 Å (Fig. 3a[12–14]). Radiation damage should be more severe for structural information at higher resolutions, but this is not the case here (cf. Fig. 2b, c). Particularly the experimental density (blue nets) for the side chain terminus of Glu 17 is missing in the 100–2.5 Å map (Fig. 2b) but recovered in the 2.5–1.19 Å map (the rightmost of Fig. 2c). The changes in negative densities with different resolution cuts were confirmed by a simulation with assigning partial charge to the terminal oxygen atoms of Asp 171 and Glu 17 (Fig. 3b; Fig. 1 of Yonekura et al.[12]). Thus, the negative densities in the weighted difference maps likely represent signals related to negative charges.

We then investigated a dose dependence of the negative densities observed above and calculated weighted-difference maps from half maps comprising first frames 1–2 (named Frame 2), 1–3 (Frame 3), and 1–20 (Frame 20) in the movie stacks (Fig. 4). The data was limited to 3.0 Å resolution, as an inclusion to 2.5 Å appeared noisy for Frames 2 and 3. The corresponding negative densities at the same density level do not decrease in the less

dose-exposed data (Frames 2 and 3 in Fig. 4), which excludes the possibility of notable radiation damage effects. There must be more radiation damage when including the later frames, but the dose-weighted summation of movie frames[23,24], which reduces the weights of higher-resolution data from later frames, probably compensated for the effect of the damage on the summed data, Frames 20 and 40.

Therefore, the observed negative densities in Figs. 1d, 2, and 4 seem to reflect the charged state of the amino acids. The above analysis is still qualitative. Difference maps could reveal smaller differences between the experimental data and model, but our recent 3D ED study shows that interpretation of residual densities in a difference map is more challenging due to distributions of partial charges and suboptimal assignment of electron scattering factors to the corresponding model[11]. Conversion to electron density[11,15,16] may provide a more interpretable result[11]. Still, further work is needed for a more extensive analysis of charges.

**Bond types of hydrogen**. In single-particle reconstructions and Coulomb potential maps by 3D ED, hydrogen densities appear further away from their parent atoms than those in X-ray crystal structures[3,10,11]. This reflects the fact that incident electrons are affected by both electrons and nuclear charges, and the latter are dominant, while X-rays are scattered by electrons around atoms, and the electron in a hydrogen atom is attracted toward its parent bonded atom[25,26]. The electron beam makes the hydrogen densities appear even slightly further from the bonded atom than the nucleus of hydrogen[3,11]. We have shown that in the Coulomb potential map of an organic molecule, the hydrogen density peak

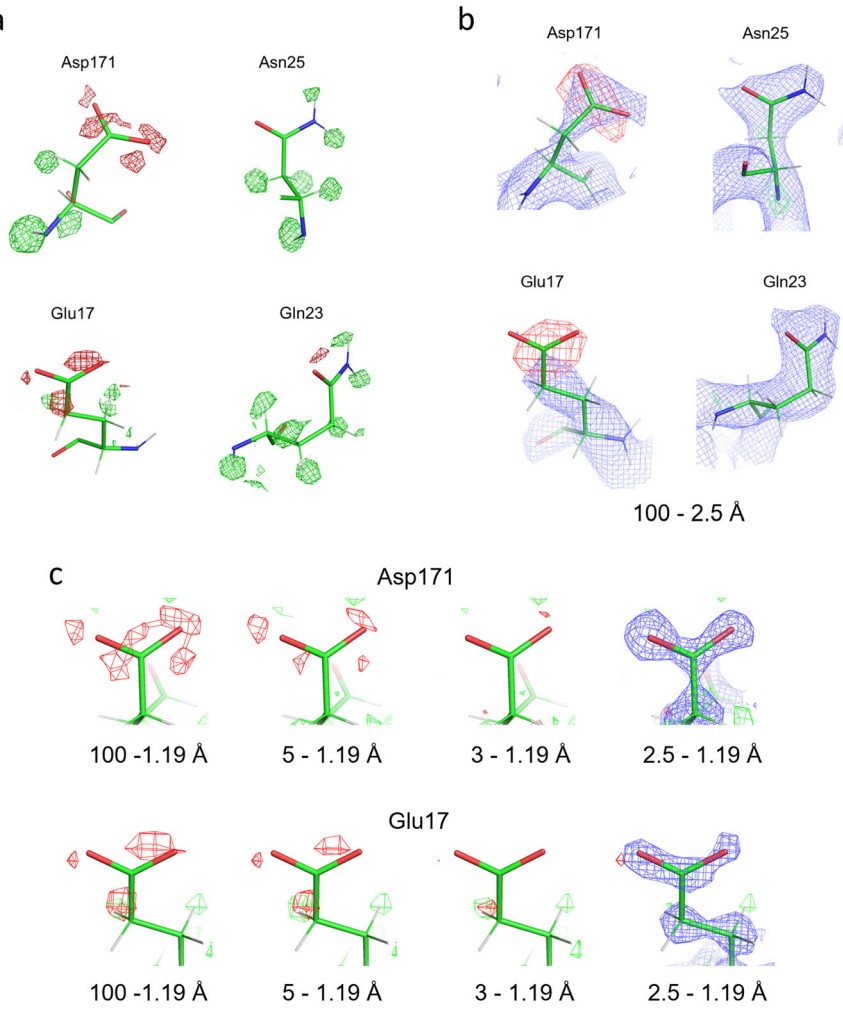

**Fig. 2 Analysis of negative densities. a** The same as in Fig. 1c but overlaid with positive and negative densities in the weighted difference map. B-factors for the terminal oxygen and bonded carbon atoms are: 23.3 Å$^2$ (OD1), 20.2 Å$^2$ (OD2), and 16.3 Å$^2$ (CG) in Asp 171; 34.0 Å$^2$ (OE1), 31.9 Å$^2$ (OE2), and 21.7 Å$^2$ (CD) in Glu 17; 19.2 Å$^2$ (OD1) and 13.2 Å$^2$ (CG) in Asn 25; and 24.6 Å$^2$ (OE1) and 15.8 Å$^2$ (CD) in Gln 23. **b** The same as in Fig. 1d (excluding the data in higher than 2.5 Å resolutions) but with the experimental map displayed in blue and at 2σ. **c** Weighted difference maps around side chains of Asp 171 and Glu 17. Calculated from the inclusion of data in given resolution ranges. Green and red nets correspond to different densities at levels of 4σ and −4σ, respectively. Experimental densities in blue are overlaid at 2σ only for those calculated from 2.5 to 1.19 Å resolution data.

position depends on the type of covalent bond[11]. There is a larger change in the position of the hydrogen peaks in electron density maps depending on the polarity of bonding due to the mobility of electrons[25–27]. In general, a higher attractive force shortens the bond length[11,25], while conversely, higher B-factors and lower resolution lengthen the apparent distance[3].

We summarize the statistics on distances of hydrogen peaks from parent atoms in the 1.19 Å cryo-EM map in Table 1. The averaged distances at density levels ≥4σ are almost identical to the nuclear positions for all bond types except for O-H bonds. The figure for O-H bonds is probably too small for a reliable estimate of the distance. Hydrogen peaks in polar bonds (N-H) appear shorter than those in other C–H$_n$ bonds. This matches well with the results obtained by 3D ED[11]. For X-rays, the peak distance in the aromatic C–H bond (C$_{aro}$–H) is shorter than that in the alkyl C–H bond (C$_{alk}$–H) due to a higher polarity of C$_{aro}$-H, whereas the nuclear positions hardly change between the two groups. We found little difference in peak distances between C$_{alk}$–H and C$_{aro}$–H in the cryo-EM map. These observations are explained by the dominance of nuclear charges in electron scattering.

The peak distances at density levels ≥2σ are longer by 0.1–0.3 Å than nuclear locations. We then plotted the hydrogen peak

position vs. B-factor and the peak position vs. peak height for each bond type (Supplementary Figs. 6 and 7). Here, B-factors along the horizontal axis of the plots are adopted from the parent non-hydrogen atoms. Although distributions are noisy and numbers in C$_{aro}$-H and O–H are small, there are weak but consistent trends showing that peak locations become longer for lower peak heights. Plots appear to be more dispersed for B-factors, which probably reflects motions or stretching vibrations of hydrogen atoms.

Lastly, we estimated the standard uncertainties in the bond lengths. A newly introduced metric RMSD$_{1/2}$ yields a value to 0.05–0.07 Å for this (see Supplementary Discussion, Supplementary Fig. 8 and Supplementary Table 2). Differences between the average lengths of N–H and C–H$_n$ bonds are at 1–2σ level (Table 1). Therefore, the error level supports the observed dependence of the peak position on the bonded-atom type.

## Conclusion
In this report, we measured charges and chemical bond types of hydrogen using single-particle cryo-EM. Electrons show a relatively higher scattering power for hydrogen over X-rays, and

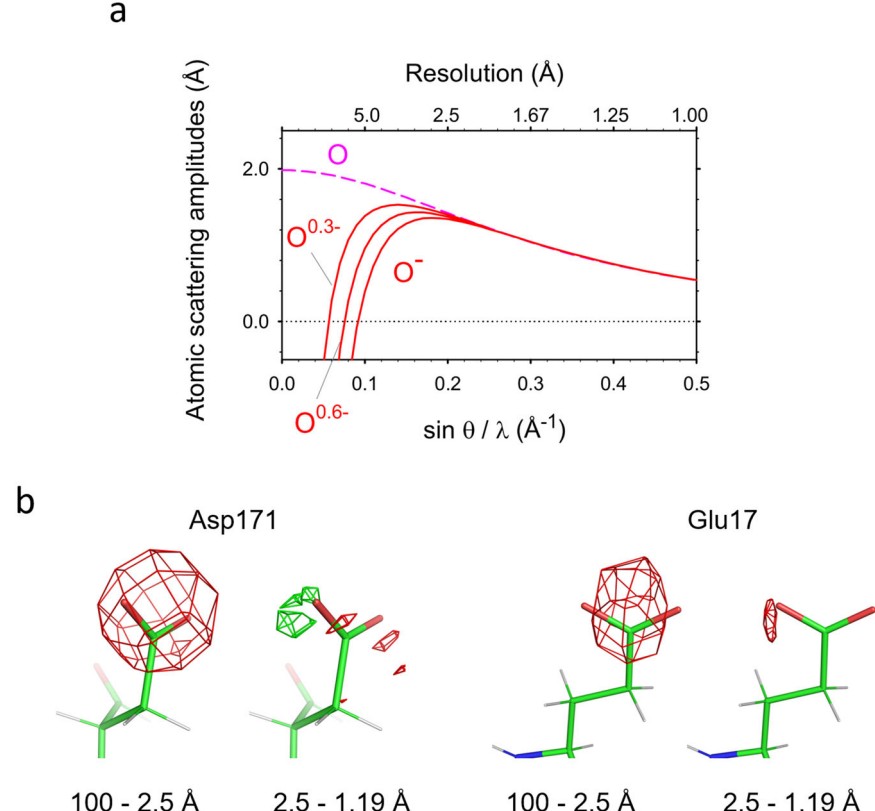

**Fig. 3 Theoretical consideration of negative charges on the oxygen atom. a** Electron scattering factors of oxygen[13,14]. Plots of partially charged $O^{0.3-}$ and $O^{0.6-}$ were calculated from a linear combination of those of the neutral and fully ionized oxygen atoms[13]. **b** Simulated Fourier difference maps by subtracting model structure factors consisting of all neutral atoms from those with assigning partial charge of $-0.3$ to only the terminal oxygen atoms of Asp 171 and Glu 17. Shown at the same display level. The modeled structures were converted to the structure factors by SFALL from the CCP4 suite[37]. Different fractions of charges do not change the appearance of difference maps much, and here $-0.3$ was adopted from Yonekura & Maki-Yonekura[13].

electron scattering factors at low-resolution further increase for positively charged atoms[12–14]. SPA also benefits from phase information derived from real images. Phase errors introduced by phase flipping and beam tilts from the axial coma-free axis can be estimated and minimized by iterative refinement processes[28]. Thereby, the sub-1.2 Å cryo-EM map has the accuracy in distinguishing the average hydrogen positions depending on the chemical bond type or the bond polarity. Moreover, weighted difference maps by the selection of spatial resolutions and from a dose frame series reveal signals likely related to negative charges on acidic-amino acids. Thus, this study demonstrates the potential of high-resolution single-particle cryo-EM, and the technique will become a powerful tool for analyzing chemical bond properties and charges.

## Methods

**Sample preparation**. Apoferritin was expressed in *Escherichia coli* BL21-Gold(DE3) (Agilent) from a plasmid encoding mouse heavy-chain ferritin provided by Dr. Haruaki Yanagisawa, the University of Tokyo. The protein was purified by heat treatment at 70 °C for 10 min, precipitation with ammonium sulfate, followed by gel filtration as described in the attached document. The apoferritin complex of ~500 kDa was eluted in 20 mM HEPES pH 7.5 and 300 mM NaCl, and the protein concentration was measured with the BCA assay (Pierce). Three μl of a sample solution containing apoferritin at a concentration of 3 or 4 mg ml$^{-1}$ was applied onto a holey carbon-coated grid (Quantifoil R1.2/1.3, Quantifoil Micro Tools GmbH) with 200 mesh, and the grid was blotted off with filter paper for 4 s and immediately plunge-frozen in liquid ethane using an FEI Vitrobot Mark IV (ThermoFisher Scientific) under 100% humidity at 4 °C.

**Adjustment of compensation for beam tilts and shifts**. The spot size, illumination angle α, and magnification were adjusted for the data acquisition condition. The objective lenses were reset to the standard focus, and the sample height ($z$) was

brought to the just-focus position. The beam-tilt compensator (condenser lens deflector tilt; CL Comp Tilt) was first adjusted as per instructions in the manual provided by JEOL. Next, the illumination was set to the parallel condition, and an object was centered and defocused to $-10$ μm with no condenser aperture inserted. Then, $x$ and $y$ beam shifts were changed by positive or negative adjustments a few μm at a time, and the overall image movement was minimized with the beam-shift compensator (condenser lens deflector shift; CL Comp Shift). ChkLensDef in ParallEM[1] can store and restore CL Comp Tilt and Shift for each illumination angle $\alpha$[4].

**Data collection**. The samples were examined at a specimen temperature of ~93 K with a CRYO ARM 300 electron microscope (JEOL) equipped with a cold-field emission gun and operated at an accelerating voltage of 300 kV. A parallel electron beam illuminated the sample, and inelastically scattered electrons were removed through an in-column energy filter with an energy slit width of 20 eV. Dose-fractionated images were recorded on a K3 camera (Gatan, AMETEK) in super-resolution mode with a nominal magnification of 100,000×, which corresponded to a physical pixel size of 0.495 Å.

All image data were acquired using SerialEM[29], and two datasets, Datasets A and B, were collected with and without JAFIS Tool (JEOL UK) ver. 1, respectively. JAFIS Tool was developed by Dr. Bartosz Marzec, JEOL UK, and can be called from the SerialEM script. It synchronizes image shifts with beam tilts and objective lens stigmas for the removal of axial coma aberrations and twofold astigmatism based on calibration, which can be prepared through a user-friendly GUI tool. The axial coma-free alignment was done as above for Dataset A before setting up JAFIS Tool. The Multiple Record setups and coma-free calibration in SerialEM were used for Dataset B.

Once the stage was aligned to a new hole by cross-correlation with the reference hole image taken at ×8000, image shifts were applied over neighboring holes around the central hole. One image stack was recorded from each of the 5 × 5 holes (Dataset A) and 3 × 3 holes (Dataset B). The settings of lenses and deflector coils were monitored with ChkLensDef in ParallEM[1,4].

In Dataset A, a total of 12,114 image stacks was collected in CDS mode at a dose rate setting of 3.819 e$^-$ s$^{-1}$ per physical pixel and 0.0585 s per frame with a total of 2.34 s exposure and a defocus setting range of $-0.5$ to $-1.0$ μm. In Dataset B, a total of 2173 was in non-CDS mode at a dose rate setting of 15.872 e$^-$ s$^{-1}$ per

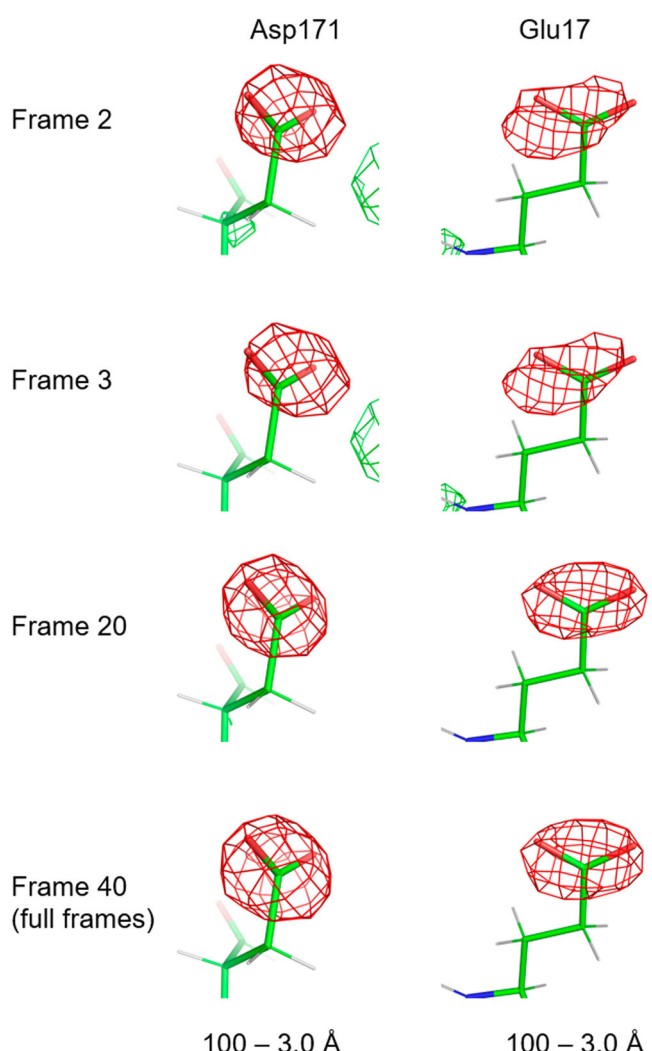

**Fig. 4 Negative densities in weighted difference maps.** Calculated from half maps comprising first frames 1–2 (Frame 2), 1–3 (Frame 3), 1–20 (Frame 20), and 1–40 (full frames; Frame 40) in the movie stacks. Total doses for Frames 2, 3, 20, and 40 correspond to ~2 e⁻ Å⁻², ~3 e⁻ Å⁻², ~20 e⁻ Å⁻², and ~40 e⁻ Å⁻², respectively. Calculated by limiting the data to 3.0 Å resolution and shown at the same absolute density levels as $-4\sigma$ in red and $4\sigma$ in green for Frame 40.

physical pixel and 0.016 s per frame with total of 0.8 s exposure and a defocus setting range of −0.5 to −1.5 μm. The interval for flashing the gun was set to 8 h. Imaging parameters are shown in Supplementary Table 1.

**Structure analysis**. We first removed micrographs taken from the regions less than 1.5 μm apart from those exposed before. This resulted from stage alignment errors in locating holes at ×8000. A Python script named distpos.py (Supplementary Data 1) was made to remove these micrographs based on $x$, $y$ coordinates in SerialEM metadata files (.mdoc) of hole images. Subsequent image processing was carried out with RELION-3.1[24,28]. Image stacks were drift-corrected, dose-weighted, and summed with a MotionCor2-like algorithm[30] implemented in RELION. CTF parameters were estimated with CTFFIND[31], and images showing few Thon rings and/or many crystal ice patterns were removed by the eye through a montage display of Fourier transforms in RELION. A 3D reconstruction of apoferritin previously obtained[4] was low-pass filtered to 20 Å, and its 2D projections were used as templates for automatic particle picking. Picked-up particles were extracted in 120 × 120-pixel boxes with a pixel size of 1.98 Å, and reference-free 2D classification was repeated twice to select ones with good class averages. In total, 2,235,864 and 311,583 particles were selected and extracted in 480 × 480-pixel boxes with a pixel size of 0.495 Å from 7852 and 1122 micrographs for Datasets A and B, respectively, and applied to 3D auto-refinement with octahedral symmetry enforcement. Anisotropic magnification, beam tilts, and CTF parameters were refined for each micrograph, followed by the estimation and correction of particle-based motions. The refinement steps were iterated a few times, and 3D maps were

reconstructed with the correction of the Ewald-sphere curvature. The resolutions of the maps were estimated to be 1.21 Å and 1.49 Å for Datasets A and B, respectively, by the gold-standard the gold standard Fourier shell correlation (FSC) criteria[32] after applying the standard post-processing procedure of RELION. Only particles in Dataset A were re-extracted in 600 600-pixel boxes with a pixel size of 0.396 Å, and the same refinement steps were repeated. Reconstruction with Ewald-sphere correction gave the final map at 1.19 Å resolution. Removal of micrographs with refined scale factors (rlnGroupScaleCorrection) <0.5³ and $x$, $y$ beam tilt deviations >0.15 mrad from the averages yielded 2,104,187 particles, but this improved the resolution of the reconstructed map only by order of 10⁻³ Å. The Rosenthal–Henderson plots[20] were calculated for both datasets by dividing the entire data into subgroups with various particle numbers. An Ewald-sphere correction was applied to subgroups consisting of ≥3200 particles.

An atomic model of mouse apoferritin (PDB ID: 7KOD) was fitted onto the 1.19 Å resolution map and inspected using UCSF Chimera[33]. The model was refined with ISOLDE[34] and REFMAC5[35] without hydrogen atoms. Then, unfiltered cryo-EM half maps were cut out into a box of 320 × 320 × 320 pixels. A Fourier $F_o$–$F_c$ difference map was calculated between the cut-out maps and the model omitting hydrogen atoms by *Servalcat*[5] in CCP-EM[36]. Hydrogen atoms and water molecules with a density level of ≥2σ or ≥4σ were picked out using PEAKMAX from the CCP4 suite[37]. Picked hydrogen atoms were manually selected by referring to the riding positions in the protein model. Structure refinement was carried out while retaining hydrogen atoms associated with the parent atoms with an average B-factor ≤ 20 Å² and removing all hydrogen atoms belonging to water molecules or amino acid residues showing multiple conformations. The hydrogen positions were fixed during refinement. Anisotropic and isotropic temperature factors were used for non-hydrogen atoms and hydrogen atoms, respectively. Refinement statistics of the model were calculated using the CCP-EM validation program[36] and are summarized in Supplementary Table 1. Q-score was calculated by replacing disordered side chains with alanine[38]. Structure figures were prepared with PyMOL (The PyMOL Molecular Graphics System, Schrödinger, LLC) and UCSF Chimera. For the estimation of RMSD₁/₂, restrained molecular dynamics (MD) refinement of the atomic model was performed against each half map using ISOLDE[34] independently. Then, RMSD was calculated between the two resultant models.

**Dose-dependence test**. We extracted polished particles from first frames 1–2 (named Frame 2), 1–3 (Frame 3), and 1–20 (Frame 20) in the movie stacks. The 3D structure was reconstructed from each extracted dataset using the refined alignment parameters for the full frames. The resolutions of the structures were estimated to be 1.37, 1.37, and 1.19 Å for Frames 2, 3, and 20, respectively, and weighted difference maps were calculated for these frame series as described above.

**Table 1 Distance between the parent atom and hydrogen density peak or nucleus or electron in the hydrogen atom.**

| Bond type[a] | Distance in Å | | Hydrogen nucleus[b] | Hydrogen electron[c] |
|---|---|---|---|---|
| | hydrogen peak | | | |
| | ≥2σ | ≥4σ | | |
| C–H[d] | 1.15 (0.13)[g] 228[h] | 1.09 (0.08) 43 | – | – |
| C_alk–H[e] | 1.15 (0.12) 185 | 1.09 (0.09) 39 | 1.09 | 0.97 |
| C_aro–H[f] | 1.14 (0.15) 43 | 1.10 (0.06) 4 | 1.08 | 0.93 |
| C–H₂ | 1.15 (0.12) 296 | 1.10 (0.11) 62 | 1.09 | 0.97 |
| C–H₃ | 1.17 (0.09) 181 | 1.09 (0.06) 28 | 1.09 | 0.97 |
| N–H[i] | 1.06 (0.09) 187 | 1.03 (0.07) 83 | 1.02 | 0.86 |
| O–H | 1.04 (0.07) 13 | 1.07 (0.10) 3 | 0.98 | 0.84 |

[a]Only for hydrogen densities detected in this structure excluding water.
[b]Derived by neutron crystallography.
[c]Derived by X-ray crystallography[39].
[d]Both the alkyl and aromatic groups.
[e]The alkyl group.
[f]The aromatic groups.
[g]The standard deviations are in parentheses.
[h]Number of observed peaks at the lower row of the cell.
[i]Peptide bonds and side chains of asparagine, glutamine, histidine, and arginine. Excluding lysine side chains due to facing solvent.

**Reporting summary**. Further information on research design is available in the Nature Portfolio Reporting Summary linked to this article.

## Data availability

A cryo-EM density map and the atomic model of apoferritin for Dataset A have been deposited in the Electron Microscopy Data Bank, and the Protein Data Bank (PDB) with accession codes EMD-35984 and 8J5A, respectively, and a cryo-EM density map for Dataset B with EMD-35981. Raw frame movies of Datasets A and B have been deposited in the Electron Microscopy Public Image Archive with accession codes EMPIAR-11534 and -11532, respectively. A Python script distpos.py is available as Supplementary Data 1.

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

## Acknowledgements

We thank Sohei Motoki, Fumiaki Makino, and Bartosz Marzec for technical support for axial-coma-free alignment and implementation of the JAFIS Tool, Haruaki Yanagisawa for providing a plasmid of mouse apoferritin, Hisashi Naitow for setting up RELION, Yuko Kageyama for help in sample preparation, and Kazuto Arakawa for information on Debye-Scherrer rings of Pt. This work was partly supported by JST-Mirai Program Grant Number JPMJMI20G5 (to K. Y.), the Cyclic Innovation for Clinical Empowerment (CiCLE) from the Japan Agency for Medical Research and Development, AMED (to K.Y.), and Research Support Project for Life Science and Drug Discovery (Basis for Supporting Innovative Drug Discovery and Life Science Research (BINDS)) from AMED under Grant Number JP22ama121006 (to K.Y.).

## Author contributions

S.M.-Y. and K.Y. conceived the project. S.M.-Y. prepared protein samples. S.M.-Y. and K.Y. collected cryo-EM data. K.Y. processed the data, and K.Y., K.T., and T.H. made codes and scripts for processing and analyzing data. K.K. built atomic models, calculated data statistics, and introduced RMSD$_{1/2}$. S.M.-Y., K.K., K.T., and K.Y. interpreted the results. K.Y. wrote the paper, and K.K. and K.Y. made figures and tables. All authors discussed the results and improved the paper.

## Competing interests
The authors declare no competing interests.
