## [Peer Review File · Communications Chemistry]

Measurement of charges and chemical bonding in a cryo-EM structureReviewers' comments:

Reviewer #1 (Remarks to the Author):

The study by Yonekura et al describes a deep study of charges in cryoEM maps. While the work is sound and I cant find many faults, I fail to see the novelty of the study. Several preprints and peer reviewed publications are already published by others in the field, some with even more complete results. It is not clear what value this work adds at this stage especially given that some calculations may be over estimated.

Reviewer #2 (Remarks to the Author):

The manuscript by Yonekura et al. has been substantially revised to address most of my suggestions. In this revision, they focused on the potency of atomic cryo-EM to study the charge states of residues and measure the accurate positions of hydrogen atoms. Their study suggests that the atomic cryo-EM could be utilized to study the fine chemical process of macromolecular complex, which would be important for the field of enzymology to enrich our knowledge of enzymatic catalytic process. There are only a few points to be considered before publication.

1. Page 4, Line 1-3. It would be good to give more explanations about the limitations of sub-atomic resolution X-ray crystallography and 3D ED to study charge distributions inside proteins.
2. Supplementary Table 1. It would be important to provide more information in this table for a fair comparison. As the authors mentioned in Methods and Supplementary Discussions, they used different beam-tilt correction strategies for Datasets A and B. These important settings for data collection need to be included in the table. In addition, the EMPIAR entries are also important to appear in this table. By the way, why did the authors not deposit structural mode of Dataset B into EMDB and PDB. This reviewer thought the deposition of Dataset B would be also useful for the community to make a comparison with Dataset A in the future.
3. Legend of Supplementary Figure 8. Please provide a full name of "DPI" (diffraction-component precision index) here.

Reviewer #3 (Remarks to the Author):

Overview:

The advancement of single-particle cryo-EM has enabled high-resolution structure determination of a biological molecule, demonstrating that conventional bright-field imaging could be well presented the atomic arrangements of a target object. This paper showed one more example of using apoferritin, a standard testing sample for cryo-EM, to re-examine the capability of cryo-EM in providing high-resolution structural information. However, the report did not accurately present some critical statements behind fundamental physics and chemistry. The experimental procedures and structural

analyses, particularly the data processing, seemed convoluted and were not carefully designed. Considering the current presentation of the manuscript, the new data and insight into either electron microscopy or chemistry are very limited. It is strongly advised to incorporate accurate interpretation and appropriate data processing procedures to provide a robust and meaningful conclusion.

Major:

1. A cryo-EM reconstruction that was built by the intensity variations from a bright-field beam can only approximate the charge information and does not provide actual phases of the target object. The statement made by the authors was too strong but not essentially true. In bright-field imaging, the azimuthal phases used for 3D reconstruction are not linearly proportional to the inner potential of the target object and can be modulated by many factors, such as defocus and lenses aberration. In addition, the 3D reconstruction is built upon multiple averaging during back-projection, relying on accurate Euler angular assignment. The authors may need to use the terms and make the statements conservatively. On the other hand, the charge information can be directly obtained from electron interference, generated by holographic imaging or CBED.
2. The reported resolution is better than the previous apoferritin study, but it does not create new insights into relevant fields, such as electron microscopy, chemistry, or biology. If this creates a new avenue to generate knowledge in chemistry, the authors may present more than one case to show the method or analysis is feasible but not only the standard testing sample.

Minor:

1. Conclusion (page 10, line 4): The SPA does not provide “accurate” phase information derived from an image, but the object phases can be “approximated” from the image intensities, the product of the exit wave with its conjugate. In addition, the phases obtained from the TEM imaging are notoriously modulated by CTF, and the bright-field TEM does not seem to be a good method for an “accurate” phase measurement.
2. Conclusion (page 10, line 5): The phase flipping is based on the predicted model CTF, which modulates the amplitudes due to spatial and temporal incoherence. In addition, it is not clear about “... and the refinement process corrects for this”.
3. Method – Adjust of compensation for beam tilts and shifts (page 17, paragraph 1, line 2): The “objective” lenses, not “object” lenses.
4. Method – Data collection (page 18, paragraph 2, line 2): The JAFIS Tool will need a citation; if not, the authors may prepare a plan to deposit the scripts in the public domain if possible, which needs to mention in the manuscript.
5. Method – Structural analysis: The authors may need to be certain that the stage alignment errors did not influence the grid mapping; if they were correlated, using the coordinates from the SerialEM (.mdoc) file to curate the micrographs that were closely imaged may not be appropriate.
6. Method – Structural analysis (line 5): MotionCor2 algorithm was not implemented in Relion. It was a standalone program, wrapped in Relion.
7. Method – Structural analysis (line 7): What are the criteria of curating the data by judging a “poor” Thon ring? Was it the number of the Thon ring or the sphericity of the Thon ring or perhaps others?
8. Method – Structural analysis (line 8): It will be better to generate a 3D model directly from your data, which presents its own signal characteristics. It seemed uncertain why the authors would choose the

model from other datasets to be the reference for structural refinement.

9. Method – Structural analysis (page 20, line 6): The resolution of a single-particle reconstruction is usually determined by the gold-standard FSC criteria, not “the standard post-processing procedure of RELION”.

10. Method – Structural analysis (page 20, line 9): When removing micrographs with refined scale factors < 0.5 , were the scaling factors for individual micrographs or images re-calculated?

11. Supplementary Figure 1: According to the numbers given, the corresponding dose rate would be about $251 \text{ e}/\text{\AA}^2/\text{second}$, accumulating for 15 seconds to a total dosage of $3,765 \text{ e}/\text{\AA}^2$. Although it is acceptable that the CFE provides high coherence and intensity for imaging, such an experiment might not demonstrate whether the performance will be comparable when using low-dose conditions ($\sim 30 \text{ e}/\text{\AA}^2$).

12. Supplementary Discussion 1: It would be better for the authors to provide the experimental conditions, such as beam tilt, for the readers to know or calculate how large the phase errors were generated. Also, the processing procedure during single-particle reconstruction utilizes averaging, including symmetrization, that mitigates the defects led by imaging (Cheng et al., 2018; Glaeser et al., 2011). Therefore, the resolution of a single-particle reconstruction cannot be fully utilized as one criterion accounted for the beam-tilt experiment.

13. Supplementary Discussion 2: The author may need to consider the scaling of magnification during the imaging process. Using half-maps from the same dataset does not provide an accurate scaling in magnification. One can see the intra-variance with the two half-maps, but not any valuable information for absolutely accurate atomic positions.

Responses

To Reviewer #1:

> The study by Yonekura et al describes a deep study of charges in cryoEM maps. While the work is sound and I cant find many faults, I fail to see the novelty of the study. Several preprints and peer reviewed publications are already published by others in the field, some with even more complete results. It is not clear what value this work adds at this stage especially given that some calculations may be over estimated.

We carefully conducted multiple tests, which support our observations of signals related to charges in a high-resolution single-particle cryo-EM structure (Figs. 1 – 4) for the first time. We also measured the dependence of average hydrogen positions on the bond type or the polarity of bonding in the cryo-EM map. We, however, do not say that hydrogen positions are a novelty in this manuscript. What we would like to show here and share with readers is whether these differences could be experimentally distinguished and treated as observable by single-particle cryo-EM. This is a major issue that the previous studies (Yip et al., *Nature*, 2020; Nakane et al., *Nature*, 2020; Yamashita et al., *Acta Cryst. D*, 2021) did not answer.

Such information decides the chemical environments and properties of proteins and should be important for not only chemists but also many researchers. We believe this is a significant advance in single-particle cryo-EM. We corrected “Introduction” to clarify the significance and aim of this study.

To Reviewer #2:

> The manuscript by Yonekura et al. has been substantially revised to address

most of my suggestions. In this revision, they focused on the potency of atomic cryo-EM to study the charge states of residues and measure the accurate positions of hydrogen atoms. Their study suggests that the atomic cryo-EM could be utilized to study the fine chemical process of macromolecular complex, which would be important for the field of enzymology to enrich our knowledge of enzymatic catalytic process. There are only a few points to be considered before publication.

We thank you so much for your positive evaluation of our revision.

> 1. Page 4, Line 1-3. It would be good to give more explanations about the limitations of sub-atomic resolution X-ray crystallography and 3D ED to study charge distributions inside proteins.

Thank you for this comment. We corrected the text as,

“But, studies of charge distributions inside proteins have been done mainly by ultra-high resolution X-ray crystallography (17, 18, 19) that are limited to samples with superb crystallinity. The application of 3D ED still has unsolved tasks such as attributions of partial charges and treatment of charged scattering factors (11-14)”.

Please also see our revisions in “Introduction”.

> 2. Supplementary Table 1. It would be important to provide more information in this table for a fair comparison. As the authors mentioned in Methods and Supplementary Discussions, they used different beam-tilt correction strategies for Datasets A and B. These important settings for data collection need to be included in the table. In addition, the EMPIAR entries are also important to appear in this

table. By the way, why did the authors not deposit structural mode of Dataset B into EMDB and PDB. This reviewer thought the deposition of Dataset B would be also useful for the community to make a comparison with Dataset A in the future.

Thank you for this suggestion. We added the beam-tilt and -shift compensation schemes, data collection schemes, EMPIAR entries, and other details in Supplementary Table 1. We deposit the cryo-EM map of Dataset B to EMDB but did not build an atomic model for Dataset B, as a single-particle cryo-EM map of apoferritin at 1.49 Å resolution has little relation to the major topics of this manuscript.

> 3. Legend of Supplementary Figure 8. Please provide a full name of “DPI” (diffraction-component precision index) here.

Corrected.

To Reviewer #3:

Overview:

> The advancement of single-particle cryo-EM has enabled high-resolution structure determination of a biological molecule, demonstrating that conventional bright-field imaging could be well presented the atomic arrangements of a target object. This paper showed one more example of using apoferritin, a standard testing sample for cryo-EM, to re-examine the capability of cryo-EM in providing high-resolution structural information. However, the report did not accurately present some critical statements behind fundamental physics and chemistry. The experimental procedures and structural analyses, particularly the data processing, seemed convoluted and were not carefully designed. Considering the current

presentation of the manuscript, the new data and insight into either electron microscopy or chemistry are very limited. It is strongly advised to incorporate accurate interpretation and appropriate data processing procedures to provide a robust and meaningful conclusion.

Thank you for evaluating the manuscript. For experimental procedures, structural analyses, and data processing, we thoroughly revised the corresponding parts according to the referees' comments and suggestions. Please see our responses below for these parts.

For interpretation of our observations of signals related to charges, we carefully designed and carried out examinations as described in the text. The side chains of aspartate and glutamate are known to be particularly susceptible to radiation. The weighted difference map calculated by Servalcat shows large negative densities appear at these side chains when removing the data at resolutions higher than 2.5 Å (Fig. 1d and Fig. 2b). In maps including of high-resolution terms, such negative densities gradually decrease as lower-resolution data are omitted (Fig. 2c). These observations are consistent with the shapes of charged electron scattering factor curves in Fig. 3a and a simulation with assigning partial charge shows also consistent results (Fig. 3b; Fig. 1 of ref. 12). Radiation damage should be more severe for structural information at higher resolutions, but this is not the case here. Furthermore, large negative densities in difference maps calculated to 3.0 Å resolution are little changes upon dose series (Fig. 4), excluding the possibility of notable radiation damage effects. Thus, we think our observations are related to charges. We would appreciate it if you could indicate what parts of these interpretations are still inappropriate. Please also refer to our responses above and below.

Major:

> 1. A cryo-EM reconstruction that was built by the intensity variations from a bright-field beam can only approximate the charge information and does not provide actual phases of the target object. The statement made by the authors was too strong but not essentially true. In bright-field imaging, the azimuthal phases used for 3D reconstruction are not linearly proportional to the inner potential of the target object and can be modulated by many factors, such as defocus and lenses aberration. In addition, the 3D reconstruction is built upon multiple averaging during back-projection, relying on accurate Euler angular assignment. The authors may need to use the terms and make the statements conservatively. On the other hand, the charge information can be directly obtained from electron interference, generated by holographic imaging or CBED.

Thank you for this comment. We agree that single-particle reconstructions that are based on bright-field imaging by electron microscopy, are affected by many factors compared to other techniques. Progresses in hardware, theories, and software have reduced these difficulties, and we believe that single-particle cryo-EM now could be applied to the measurement of more detailed properties in biological macromolecules. So, we carefully conducted multiple tests by spatial-resolution selection (Figs. 1 and 2), a dose-dependent frame series (Fig. 4), and theoretical calculations (Fig. 3). These tests support our observation of the signals related to charges, as in our response above.

Other techniques, such as holographic imaging, CBED, 3D ED, and high-resolution XRD used for analyses of Coulomb potentials and/or charge density, cannot reach detailed structures of protein molecules dispersed in amorphous ice. So, we think it is important to show that charges and chemical bond types in these targets can be observed

also with this method, which can expand application targets for the investigation of experimental Coulomb potentials. We modified “Introduction” to clarify the significance and aim of this study as,

“This feature has been exploited to measure charge distributions in covalent crystals of inorganic materials from convergent-beam electron diffraction (CBED) patterns (e.g. 15, 16) and to investigate charged states at functional sites in thin crystals of proteins (12) and small organic compounds (11) by rotation 3D electron diffraction with a parallel beam (3D ED). However, different from those techniques, single-particle cryo-EM by bright-field electron imaging is affected by many factors. These include phase errors caused by intensity variations and beam tilts from the axial coma-free axis, modification and decay of amplitudes by the contrast transfer function (CTF), and errors in Euler angular assignment for 3D reconstruction. Progresses in hardware, theories, and software have reduced these difficulties, and we believe that single-particle cryo-EM now could have the potential to measure more detailed properties including information related to charges. This will expand application targets for the investigation of experimental Coulomb potentials. Of course, charges are critically important for the function of macromolecules. But, studies of charge distributions inside proteins have been done mainly by ultra-high resolution X-ray crystallography (17, 18, 19) that are limited to samples with superb crystallinity. The application of 3D ED still has unsolved tasks such as attributions of partial charges and treatment of charged scattering factors (11-14)”.

We carefully revised the manuscript by using conservative terms and statements and avoiding strong expressions.

> 2. The reported resolution is better than the previous apoferritin study, but it

does not create new insights into relevant fields, such as electron microscopy, chemistry, or biology. If this creates a new avenue to generate knowledge in chemistry, the authors may present more than one case to show the method or analysis is feasible but not only the standard testing sample.

Thank you for this comment. This manuscript reports the first observation of signals related to charges in a high-resolution single-particle cryo-EM structure. As mentioned, we carefully conducted multiple tests (Figs. 1 - 4), which support our observation. Of course, charges are critically important for the stabilization of protein structures and their functions, but studies of charge distributions inside proteins are limited. This study also measured the dependence of average hydrogen positions on the bond type or the polarity of bonding in the cryo-EM map for the first time. We, however, do not say that hydrogen positions are a novelty in this manuscript. What we would like to show here and share with readers is whether these differences could be experimentally distinguished and treated as observable by single-particle cryo-EM. This is a major issue that the previous studies (Yip et al., *Nature*, 2020; Nakane et al., *Nature*, 2020; Yamashita et al., *Acta Cryst. D*, 2021) did not answer. Such information decides the chemical environments and properties of proteins and should be important for not only chemists but also many researchers.

We agree that more demonstrative studies from different research groups would be required for broader applications. We think our work is the first one for this and demonstration of other examples would be beyond the scope of this research. Instead, this study could provide the next standards of expected information such as signals related to charges, how to distinguish these signals and radiation damage, bond properties of hydrogen, and the accuracy and reliability of the data, from high-resolution single-particle

cryo-EM. Such information and our approaches could be applied to analyses of other high-resolution structures and so have high generalizability. We believe this is a significant advance in single-particle cryo-EM.

Minor:

> 1. Conclusion (page 10, line 4): The SPA does not provide “accurate” phase information derived from an image, but the object phases can be “approximated” from the image intensities, the product of the exit wave with its conjugate. In addition, the phases obtained from the TEM imaging are notoriously modulated by CTF, and the bright-field TEM does not seem to be a good method for an “accurate” phase measurement.

We agree this expression is too strong, and corrected the sentence as,
“SPA also benefits from phase information derived from real images”.

> 2. Conclusion (page 10, line 5): The phase flipping is based on the predicted model CTF, which modulates the amplitudes due to spatial and temporal incoherence. In addition, it is not clear about “... and the refinement process corrects for this”.

Thank you for this comment. We corrected this part as,

“Phase errors introduced by phase flipping and beam tilts from the axial coma-free axis can be estimated and minimized by iterative refinement processes”.

> 3. Method – Adjust of compensation for beam tilts and shifts (page 17, paragraph 1, line 2): The “objective” lenses, not “object” lenses.

Corrected. Thank you for this comment.

> 4. Method – Data collection (page 18, paragraph 2, line 2): The JAFIS Tool will need a citation; if not, the authors may prepare a plan to deposit the scripts in the public domain if possible, which needs to mention in the manuscript.

JAFIS Tool is sold by JEOL UK now. We corrected the text.

> 5. Method – Structural analysis: The authors may need to be certain that the stage alignment errors did not influence the grid mapping; if they were correlated, using the coordinates from the SerialEM (.mdoc) file to curate the micrographs that were closely imaged may not be appropriate.

Exposed positions were all determined by aligning to a hole in an image taken at $\times 8000$, and the removal of micrographs was based on these positions. So, mapping a whole grid at $\times 50$ or capturing square images at $\times 150$ had no influence on the “curation”.

To clarify this, we corrected the text as,

“We first removed micrographs taken from the regions less than 1.5 μm apart from those exposed before. This resulted from stage alignment errors to locate holes at $\times 8000$. A python script named distpos.py (Supplementary code) was made to remove these micrographs based on x, y coordinates in SerialEM meta data files (.mdoc) of hole images”.

> 6. Method – Structural analysis (line 5): MotionCor2 algorithm was not implemented in Relion. It was a standalone program, wrapped in Relion.

A MotionCor2-like algorithm is implemented as described in Zivanov et al.,

IUCrJ 6, 5–17 (2019). This can be confirmed from the GUI of RELION as below, and we corrected the corresponding word.

> 7. Method – Structural analysis (line 7): What are the criteria of curating the data by judging a “poor” Thon ring? Was it the number of the Thon ring or the sphericity of the Thon ring or perhaps others?

We removed micrographs showing few Thon rings and/or many crystal ice patterns. This inspection can be easily done through montage of Fourier transforms in a display of RELION. We corrected the text as,

“... images showing few Thon rings and/or many crystal ice patterns were removed by eye through a montage display of Fourier transforms in RELION”.

> 8. Method – Structural analysis (line 8): It will be better to generate a 3D model directly from your data, which presents its own signal characteristics. It seemed uncertain why the authors would choose the model from other datasets to be the

reference for structural refinement.

The structure refinement was started using a reference map low pass-filtered to 20 Å resolution and iterated to the final resolutions beyond 1.5 Å. The final reconstructions less depend on the initial reference map.

> 9. Method – Structural analysis (page 20, line 6): The resolution of a single-particle reconstruction is usually determined by the gold-standard FSC criteria, not “the standard post-processing procedure of RELION”.

Thank you for this comment. We corrected the text as,
“The resolutions of the maps were estimated to be 1.21 Å and 1.49 Å for Datasets A and B, respectively, by the gold-standard Fourier shell correlation (FSC) criteria (44) after applying the standard post-processing procedure of RELION”.

> 10. Method – Structural analysis (page 20, line 9): When removing micrographs with refined scale factors < 0.5, were the scaling factors for individual micrographs or images re-calculated?

This is for individual micrographs. Every iteration of the refinement of RELION gives scale factors for individual micrographs. We added the reference here as,
“Removal of micrographs with refined scale factors (rlnGroupScaleCorrection) < 0.5 (3) ...”.

> 11. Supplementary Figure 1: According to the numbers given, the corresponding dose rate would be about 251 e-/Å²/second, accumulating for 15 seconds to a total dosage of 3,765 e-/Å². Although it is acceptable that the CFE

provides high coherence and intensity for imaging, such an experiment might not demonstrate whether the performance will be comparable when using low-dose conditions ($\sim 30 \text{ e}/\text{\AA}^2$).

The high dose was used for taking an image of thin metal sputtered on the carbon film. We showed this figure just for demonstration of the CFE gun. The doses used for apoferritin images are less and shown in Supplementary Table 1. Although individual images of protein did not show clear signals even at 1.2 Å resolution, averaging recovered structure details to this resolution range. According to Supplementary Fig. 1, it may indicate much higher-resolution signals will be able to be recovered, but not with the Schottky emission, which does not show such high-resolution signals even from the thin metal image. Please also see a comparison of the CFE and Schottky emission in Fig. 2 of Hamaguchi et al., *J. Struct. Biol.* (2019).

> 12. Supplementary Discussion 1: It would be better for the authors to provide the experimental conditions, such as beam tilt, for the readers to know or calculate how large the phase errors were generated. Also, the processing procedure during single-particle reconstruction utilizes averaging, including symmetrization, that mitigates the defects led by imaging (Cheng et al., 2018; Glaeser et al., 2011). Therefore, the resolution of a single-particle reconstruction cannot be fully utilized as one criterion accounted for the beam-tilt experiment.

Thank you for this comment. We show beam fluctuations during data collection and provide all the parameters needed for the calculation of phase shifts in the legend of Supplementary Fig. 2. The readers can calculate phase shifts from these data. We added the sentence below for reference.

“The phase shifts become 112° at 1.5 \AA resolution and 218° at 1.2 \AA for 300 kV and a beam tilt of 0.1 mrad”.

We agree that averaging and symmetrization could mitigate defects caused by imperfect imaging including beam tilts from the coma-free axis to some extent. Cheng et al. (*J. Struct. Biol.*, 2018) showed that imaging with beam tilts of 1.3 mrad, 5 mrad, and 10 mrad yielded single-particle reconstructions of a proteasome symmetrical complex at $\sim 3 \text{ \AA}$, $\sim 4 \text{ \AA}$, and $\sim 5 \text{ \AA}$ resolution, respectively, without post-correction of phase shifts by beam tilts. We think that the present data suggest the improvement in the resolution and its limit by the post-correction of the phase shifts. We corrected the text as,

“Averaging and symmetrization are shown to reduce phase errors caused by beam tilts off the axial coma-free axis to some extent even without post-correction of phase shifts by beam tilts (46, 47). The post-correction can further improve the resolution (28). ... Thus, the present data would suggest the post-correction is probably effective to this resolution range but beyond this range becomes less so with such large beam tilt fluctuations”, where refs. 46 and 47 are Glaeser et al., *J. Struct Biol.* (2011) and Cheng et al., *J. Struct Biol.* (2018), respectively.

> 13. Supplementary Discussion 2: The author may need to consider the scaling of magnification during the imaging process. Using half-maps from the same dataset does not provide an accurate scaling in magnification. One can see the intra-variance with the two half-maps, but not any valuable information for absolutely accurate atomic positions.

Thank you for this comment. It is hard to obtain absolutely accurate atomic positions from any experimental data, but we will be able to evaluate the standard

uncertainty of atomic positions from experimental data. Half maps are shown to contain information corresponding to F_{obs} and $\sigma(F_{\text{obs}})$ of X-ray crystallography (Murshudov, *Methods Enzymol.*, 2016; Nicholls et al., *Acta Cryst. D*, 2018; Yamashita et al., *Acta Cryst. D*, 2021). Calculation of DPI in X-ray crystallography of course relies on F_{obs} and $\sigma(F_{\text{obs}})$. Thus, we think half maps would be able to be used to estimate the standard uncertainty in atomic positions. Indeed, plots of DPI in X-ray crystallography and $\text{RMSD}_{1/2}$ in single-particle cryo-EM show consistency over different samples and resolutions (Supplementary Fig. 8). We think the calculation of the half maps should follow the standard protocol without rescaling here. We added the sentence of the half maps, F_{obs} , and $\sigma(F_{\text{obs}})$ in the text.

REVIEWERS' COMMENTS:

Reviewer #3 (Remarks to the Author):

The new revision can be ready for publication.